# Dysmetabolism and Sleep Fragmentation in Obstructive Sleep Apnea Patients Run Independently of High Caffeine Consumption

**DOI:** 10.3390/nu14071382

**Published:** 2022-03-25

**Authors:** Sílvia V. Conde, Fátima O. Martins, Sara S. Dias, Paula Pinto, Cristina Bárbara, Emília C. Monteiro

**Affiliations:** 1CEDOC, NOVA Medical School, Faculdade de Ciências Médicas, Universidade Nova de Lisboa, Rua Câmara Pestana 6, Edifício 2, piso 3, 1150-082 Lisboa, Portugal; fatima.martins@nms.unl.pt (F.O.M.); emilia.monteiro@nms.unl.pt (E.C.M.); 2ciTechCare—Center for Innovative Care and Health Technology, Polytechnic of Leiria, 2411-901 Leiria, Portugal; sara.dias@ipleiria.pt; 3School of Health Sciences, Polytechnic of Leiria, 2411-901 Leiria, Portugal; 4Pneumology Department, Centro Hospitalar de Lisboa Norte, Hospital Pulido Valente, 1649-028 Lisboa, Portugal; paulagpinto@gmail.com (P.P.); cristina.barbara@chln.min-saude.pt (C.B.)

**Keywords:** caffeine, obstructive sleep apnea, apnea/hipopnea index, sleep architecture catecholamines, dysmetabolism

## Abstract

Daytime hypersomnolence, the prime feature of obstructive sleep apnea (OSA), frequently leads to high coffee consumption. Nevertheless, some clinicians ask for patients’ caffeine avoidance. Caffeinated drinks are sometimes associated with more severe OSA. However, these effects are not consensual. Here we investigated the effect of caffeine consumption on sleep architecture and apnea/hypopnea index in OSA. Also, the impact of caffeine on variables related with dysmetabolism, dyslipidemia, and sympathetic nervous system (SNS) dysfunction were investigated. A total of 65 patients diagnosed with OSA and 32 without OSA were included after given written informed consent. Polysomnographic studies were performed. Blood was collected to quantify caffeine and its metabolites in plasma and biochemical parameters. 24 h urine samples were collected for catecholamines measurement. Statistical analyses were performed by SPSS: (1) non-parametric Mann-Whitney test to compare variables between controls and OSA; (2) multivariate logistic regression testing the effect of caffeine on sets of variables in the 2 groups; and (3) Spearmans’ correlation between caffeine levels and comorbidities in patients with OSA. As expected OSA development is associated with dyslipidemia, dysmetabolism, SNS dysfunction, and sleep fragmentation. There was also a significant increase in plasma caffeine levels in the OSA group. However, the higher consumption of caffeine by OSA patients do not alter any of these associations. These results showed that there is no apparent rationale for caffeine avoidance in chronic consumers with OSA.

## 1. Introduction

Obstructive sleep apnea (OSA), the most common sleep-disordered breathing, is a highly prevalent disease [1] characterized by repetitive episodes of airflow cessation (apnea) or airflow reduction (hypopnea) during the sleep. These obstructive apneas during sleep result into recurrent arousals during sleep and in repetitive episodes of hypoxia, hypercapnia and apneas, which result in chemoreflex activation and consequent activation of the sympathetic nervous system (SNS) [2,3]. Frequent arousal can cause relevant changes on sleep architecture, namely sleep fragmentation, and may be responsible for diurnal hypersomnolence [3], as well as several adverse safety and health consequences including diurnal hypertension [4], cardiovascular disease, stroke, motor vehicle accidents, and diminished cognitive capacities and quality of life [5]. 

Daytime hypersomnolence and fatigue are regular symptoms in OSA, being frequently considered the main responsible for the high coffee consumption. It was previously described that OSA patients drink, on average, nearly three times more coffee than control subjects [6]. However, a recent meta-analysis did not show enough evidence to confirm the association between OSA and caffeine consumption [7]. Also, caffeinated drinks have been associated with more severe OSA [8] and some physicians ask for its avoidance, however these effects are not consensual as it has been found that coffee or tea do not seem to impact on OSA severity [8] and that caffeine seemed to improve cognitive function in people with OSA [9]. 

Several authors have correlated high caffeine consumption with an elevated blood pressure (BP) [10], suggesting that caffeine augments BP and total peripheral resistance in response to stressors [11,12,13]. Additionally, it has been previously described that caffeine increases SNS activity by increasing catecholamine levels, down-regulating beta-receptors [14] and blocking adenosine receptors [15]. In contrast, several studies have showed beneficial effects of long-term coffee intake on glucose metabolism and insulin action [16,17]. Moreover, some studies have associated coffee chronic consumption with a slower cognitive decline and with a reduced risk of cognitive dysfunction, dementia, and Alzheimer disease [18,19]. In particular, in moderate and severe OSA patient’s, daily caffeine intake was associated with less cognitive impairment [9]. However, despite these positive impacts of caffeine on glucose homeostasis in metabolic disease patients and on cognitive dysfunction in OSA patients [9] most of the clinicians remain asking for patient caffeine avoidance. 

Besides, all of the literature that can be found regarding the association of OSA and sympathetic activity and BP, the effect of caffeine consumption in other variables closely associated with OSA, such as sleep parameters (e.g., rapid eye movement (REM) sleep, non-REM sleep and arousals), dyslipidemia (e.g., triglycerides (TGs), cholesterol, high-density lipoproteins (HDL) and low-density lipoproteins (LDL) and dysmetabolism (e.g., body mass index (BMI) and glycemia), has not been evaluated so far. Thus, the aim of the present study was to investigate if caffeine plasma levels affect OSA severity and OSA association with dysmetabolism and sympathetic nervous system dysfunction.

## 2. Materials and Methods

### 2.1. Patients and Ethics

Eligible male patients with or without OSA (65 vs. 32) were recruited from the Respiratory Diseases Service of Hospital Pulido Valente (Lisbon). The diagnosis of OSA was established by the presence of typical clinical features of this disorder and confirmed by overnight polysomnography. Patients were included independently of their hypertension condition, hypertension ongoing medication and body mass index (BMI) and excluded if they had any psychiatric disorder, smoking habits, or inability to understand the information required for an informed consent. The study was approved by the Centro Hospitalar Lisboa Norte’s Ethics Committee (Ethical approval: 11st November 2009) and by the Ethics Committee of the NOVA Medical School and registered at ClinicalTrials.gov accessed on 17 February 2022 (NCT01803815). It was performed in accordance with the Helsinki Declaration. All volunteers gave their written informed consent.

### 2.2. Study Design

Sociodemographic and anthropometric data, comorbidities and ongoing medication profile were documented. Weight, and abdominal circumference using standardized protocols were assessed. 

Polysomnographic studies were performed using a polysomnographer (Medcare, Somnologica, Portugal). Polysomnography report included sleep parameters (e.g., sleep staging, sleep time, sleep latency, sleep efficiency, REM and non-REM sleep and arousals), respiratory events such as number and duration of apneas and hypopneas (apnea is the cessation of airflow at the nose and mouth for more than 10 s associated with oxygen desaturation of 3% and hypopnea is a discernible reduction in respiratory effort accompanied by a decrease of more than 4% of oxygen saturation) and apnea/hypopnea index (AHI). OSA was categorized according to current AHI cut-offs of less than 5 (nondiagnostic), 5≥ and <15 (mild), 15≥ and <30 (moderate), and at least 30 (severe). 

In all patients, blood and 24 h urine samples were collected, before and after polysomnographic studies and stored at −80° until analyzed. Xanthine’s concentrations, including caffeine, theobromine, theophylline and paraxanthine, glycemia total cholesterol, triglycerides (TG), HDL and LDL were measured in blood and epinephrine (E), norepinephrine (NE) and dopamine were measured in the 24 h urine. Quantifications of xanthines and catecholamines were carried by HPLC with UV and electrochemical detection, respectively, as previously described [20,21]. 

### 2.3. Statistical Analysis

Data were presented as median (interquartile range) with nonparametric Mann-Whitney test between two independent samples. Spearman correlations were performed between caffeine concentrations and sleep-related parameters in all subjects or in OSA patients and it was considered significantly correlated at *p* < 0.05. We have used non-parametric tests since variables did not follow a normal distribution.

Exploring the OSA patients we have performed an exploratory data analysis by binary logistic regression for each measured parameter for the dependent variable OSA and data were presented as odds ratio (OR), interval of confidence of 95% (IC95%) and correspondent value of p, with *p* < 0.05 being considered statistically significant. Afterwards we have grouped the significantly different variables between the 2 groups in 4 different models [dysmetabolism (glycemia and BMI), dyslipidemia (TGs, HDL and LDL), SNS (dopamine and NE) and sleep (arousals)] and tested its association with OSA when caffeine concentration in blood is present in each model. Each logistic regression was tested for goodness of fit by the Hosmer and Lemeshow test. Statistical analyses were performed using SPSS statistical software (vs 26.0 for Mac) (IBM, NY, US).

## 3. Results

Male patients diagnosed with OSA (65) and male patients without OSA (32) were recruited. The study and the control groups included volunteers aged 50 ± 8 and 34 ± 7 years, respectively. 

Table 1 summarizes findings regarding metabolic and sympathetic nervous system variables studied in OSA patients and control subjects. OSA patients showed significantly higher BMI, glycemia, total cholesterol, LDL, TG and dopamine and NE levels in comparison to control patients. OSA patients also showed a significantly lower HDL levels than control subjects (Table 1). 

In Table 2 are summarized the levels of caffeine plasmatic concentrations and sleep parameters for both groups of patients. OSA patients showed significantly higher caffeine plasmatic concentrations than control subjects. As expected, AHI and the number of arousals were also higher for patients included in the study group. As it can be observed in Table 2, no statistical differences between the REM and non-REM sleep period in patients with or without OSA were found. 

Aiming to evaluate the impact of caffeine consumption on the prediction of OSA development and on disease severity we correlated caffeine plasma levels and sleep architecture-related parameters for the OSA patients’ group. We found that there is no statistically significant correlation for any of the tested parameters, showing that caffeine is not associated with disease severity (Table 3).

To deeply understand if OSA development is associated with dyslipidemia, dysmetabolism, SNS dysfunction and/or sleep fragmentation we performed a binary logistic regression within both groups, OSA and control patients, for all the covariables significantly different in the previous nonparametric test analysis. Blood pressure was not considered as a covariable for the statistical analysis since OSA patients presented normotensive values due to the use of anti-hypertensive medication. Also, apart from this medication no relevant drugs for the present study were taken by patients. 

We found that all the tested covariables were associated with the development of OSA, showed by the *p* value in Table 4 for the differences between OR for an interval of confidence of 95% present. The binary logistic regression of each model followed the goodness of the fit by the Hosmer and Lemeshow test.

Coffee and other beverages and foods rich in caffeine consumption has been reported to be increased in OSA condition due to the sleep privation and daily sleepiness, as shown herein in the present study by the higher caffeine concentration in blood in patients diagnosed with OSA. As such, we had interest in investigating if the associations between metabolic and sleep parameters with OSA pathology are influenced by caffeine levels.

Therefore, we performed a multivariate logistic regression with OSA as dependent variable and tested the association of the pathology with dyslipidemia (TG, HDL-c, LDL-c), dysmetabolism (BMI and glycemia), SNS dysregulation (dopamine and norepinephrine) and sleep impairment (arousals) taking in account caffeine concentration in all models. Each model followed the goodness of fit represented by a non-significative Hosmer and Lemeshow test. The results for each model tested are presented in Table 5.

We found that caffeine association with OSA is lost when all the other variables related with OSA comorbidities are taken in account. Moreover, we also observed that all the variables tested maintain their association with OSA even in the presence of caffeine. Regarding the dysmetabolism model we observed that BMI and glycemia associates statistically significantly with OSA independently of the higher caffeine intake in these patients (Table 5a). Similarly, in the dyslipidemia model we found that TGs do not associate with OSA when caffeine factor is present, but HDL and LDL maintains its statistically significantly association with the pathology, with an expected protective association (OR < 1) and risk factor association (OR > 1), respectively (Table 5b). Regarding SNS function association with OSA we found that NE is statistically significantly associated with the pathology and that caffeine levels seem to have a role on SNS association with OSA (Table 5c). Finally, as expected, arousals are positively statistically significantly associated with OSA, but caffeine do not contribute for this association (Table 5d).

## 4. Discussion

In the present study, we showed that the group of OSA patients studied in the present manuscript exhibit significant dysmetabolic condition with higher BMI and glycemia, dyslipidemia with high total cholesterol, LDL, TG and lower HDL levels and deregulation of the SNS assessed as high dopamine and NE levels. Moreover, we showed, as expected, that OSA patients consume more caffeinated products to compensate daytime sleepiness, this being reflected in the high caffeine plasma levels in these patients. We also found that no correlation exists between caffeine plasma levels and OSA and its severity. Finally, we have shown that metabolic dysfunction, dyslipidemia, and sleep disturbances are associated with OSA independently of caffeine plasma levels.

Caffeine is one of the most widely consumed psychoactive substances in the World and it is known that this xanthine disrupts sleep, diminishing total sleep time and sleep onset in healthy volunteers [15,22]. Apart from some controversy on the effects of caffeine on REM sleep [23,24], it is consensual that caffeine affects arousals [25] and NREM sleep [15,26]. However, our results are not in this line of evidence since we found that caffeine plasma levels lose association with OSA when arousals are taken in account and do not correlate with any sleep-related parameters when this association is analyzed only in OSA group. When subjects from both groups are analyzed for the association of caffeine plasma levels with sleep parameters, both AHI and arousals have a statistically significant correlation, showing that OSA patients indeed drink more caffeine due to sleep fragmentation. Moreover, no correlation has been found between caffeine consumption and the severity of OSA. These results are in line with the findings of the authors in [8], showing that coffee intake is not associated with sleep disorder breathing, a wide spectrum of sleep-related conditions that includes OSA. Interestingly, and in complete contrast with the idea that caffeine may contribute to sleep disruption in OSA or even to its absence of effects, a recent study by Takabayashi et al. [27] showed in a cohort of Japanese men, that individuals who were overweight exhibited a significant inverse association between coffee consumption and oxygen desaturation index, a parameter that correlate with AHI [28,29]. While this study of Takabayashi et al. [27] did not provided causality between coffee consumption and sleep disorder breathing it clearly highlights the importance of further research on the associations between coffee consumption in sleep disorder breathing and especially in OSA. In fact, it was recently emphasized in a meta-analysis that there are insufficient data in the literature to determine whether and how, OSA is associated with caffeine consumption [7]. 

Some studies in the past have suggested that caffeinated coffee consumption was associated with a lower metabolic control and therefore with an increased risk of type 2 diabetes and metabolic syndrome [17,30], although nowadays is becoming consensual that drinking coffee (whether caffeinated and decaffeinated) may actually decrease the risk of developing metabolic syndrome and type 2 diabetes [16,31]. These results were supported by animal work showing that long-term caffeine intake prevent and reverse fat deposition, insulin resistance and glucose intolerance in hypercaloric animal models [21,32,33]. However, the results for the effect of caffeine on dysmetabolism do not seem to be extrapolated to dyslipidemia as a recent meta-analysis suggested that coffee consumption may be associated with an elevated risk for dyslipidemia [34]. Nevertheless, these results were not supported by previous clinical studies showing that coffee consumption has been positively associated with blood lipid concentration in humans [35,36] and by the findings that caffeine intake in hypercaloric animal models decrease non-esterified fatty acids [21]. Our results show that caffeine-enriched products consumption in OSA patients do not influence OSA metabolic comorbidities. Moreover, we also observed that all of the variables related with dysmetabolism and dyslipidemia tested herein, maintain their association with OSA even in the presence of caffeine, suggesting that caffeine intake in OSA is incapable of prevent or improve dysmetabolic features. Therefore, we can suggest that the mechanisms behind the development of dysmetabolism in OSA might differ from the common dysmetabolic states in where chronic caffeine intake have shown to have beneficial effects [21,32,33]. 

Regarding SNS-related parameters and following the previous findings of Bardwell et al. [6], where a positive association between NE levels and caffeine consumption in OSA patients was observed, or with the previous observations of Benowitz [14] where an increase in catecholamines circulating levels elicited by caffeine was found, herein we observed that caffeine is directly associated with NE levels in OSA. Therefore, we can suggest that higher caffeine intake in OSA patients can contribute to the overactivity of the SNS activity observed in OSA patients [37] and in here manifested by an increase in NE levels. 

In conclusion, caffeine consumption in OSA patients is not associated with OSA severity, as well as with dysmetabolism and sleep fragmentation although we cannot discharge a contribution of high caffeine levels to the overactivation of the SNS observed in OSA patients. While there is no apparent rationale for caffeine avoidance in chronic consumers with OSA, we believe that more studies regarding coffee/caffeine consumption in OSA patients should be performed. 

## Figures and Tables

**Table 1 nutrients-14-01382-t001:** Comparison between metabolic and sympathetic nervous system variables in patients with and without obstructive sleep apnea (OSA).

Variables Assessed	Patients without OSA (*n* = 32) Median (Q_1_–Q_3_)	Patients with OSA (*n* = 65) Median (Q_1_–Q_3_)
BMI (kg m^−2^)	25.15 (20.8–29.1)	32.30 (25.2–42.6) ***
Glycaemia (mg/dL)	77.00 (62.0–88.0)	88.00 (63.0–128.0) ***
Total cholesterol (mg/dL)	171.0 (118.0–237.0)	191.00 (124.0–275.0) ***
LDL (mg/dL)	98.00 (62.0–169.0)	125.00 (61.0–188.0) ***
HDL (mg/dL)	51.50 (37.0–57.0)	43.00 (27.0–61.0) ***
TG (mg/dL)	81.5 (30.0–241.0)	129.00 (59.0–417.0) *
Dopamine (μg/24 h)	236.50 (64.0–303.0)	245.96 (48.0–549.8) **
Epinephrine (μg/24 h)	4.50 (2.0–11.0)	6.0 (2.0–29.4)
Norepinephrine (μg/24 h)	30.00 (14.0–56.0)	58.0 (12.0–149.3) ***

Data are presented as median (interquartile range). LDL-c, low density lipoproteins; HDL-c, high density lipoproteins; TG, triglycerides; E, epinephrine; NE, norepinephrine (* *p* < 0.05; ** *p* < 0.01; *** *p* < 0.001, median Mann-Whitney statistic test for independent samples, corresponding to the difference between patients with and without OSA).

**Table 2 nutrients-14-01382-t002:** Comparison of caffeine and total xanthine plasmatic concentrations, Apnea-hypopnea index (AHI), sleep architecture (REM and non-REM) and arousals events in patients with or without obstructive sleep apnea (OSA).

Variables Assessed	Patients without OSA (*n* = 32) Median (Q_1_–Q_3_)	Patients with OSA (*n* = 65) Median (Q_1_–Q_3_)
Caffeine (μg/mL)	0.18 (0.0–0.77)	1.25 (0.0–13.4) **
Total xanthine (μg/mL)	0.93 (0.0–3.6)	1.91 (0.0–18.5)
AHI (events/h)	1.60 (0.2–4.9)	26.10 (6.3–137.5) ***
NREM (min)	304.50 (253.0–341.7)	291.50 (150.5–388.3)
REM (min)	44.25 (16.0–121.0)	38.50 (5.0–98.5)
Arousals (events/h)	7.00 (4.0–19.5)	30.60 (10.3–109.1) ***

Data are presented as median (interquartile range). AHI, apnea-hypopnea index; REM, rapid eye movement sleep; NREM, non-rapid eye movement sleep (** *p* < 0.01; *** *p* < 0.001, median Mann-Whitney statistic test for independent samples, corresponding to the difference between patients with and without OSA).

**Table 3 nutrients-14-01382-t003:** Correlation of caffeine concentration in OSA patients with sleep-related parameters: apnea-hypopnea index (AHI), arousals, rapid-eye movement (REM) and non-rapid eye movement (NREM).

Variables Correlated with Caffeine	OSA Patients	
	R (Spearman)	*p* Value
AHI	−0.186	0.309
Arousals	−0.164	0.370
REM	−0.050	0.791
NREM	−0.316	0.083

Data are presented as Spearman correlation coefficient between caffeine concentrations and sleep-related parameters and correspondent value of *p*, considered significantly correlated at *p* < 0.05; AHI, apnea-hypopnea index; REM, rapid eye movement; NREM, non-rapid eye movement.

**Table 4 nutrients-14-01382-t004:** Binary logistic regression for BMI, glycemia, cholesterol, LDL, HDL, TGs, dopamine, NE, caffeine concentration and arousals with OSA as dependent variable.

Dependent Variable = OSA Independent Variables	Odds Ratio (OR)	IC_95%_	*p* Value
BMI (kg m^−2^)	1.93	1.44–2.60	<0.001
Glycaemia (mg/dL)	1.22	1.11–1.34	<0.001
Total cholesterol (mg/dL)	1.03	1.02–1.05	<0.001
LDL (mg/dL)	1.04	1.02–1.05	<0.001
HDL (mg/dL)	0.92	0.87–0.96	<0.01
TG (mg/dL)	1.01	1.00–1.02	<0.05
Dopamine (μg/24 h)	1.01	1.00–1.01	<0.01
Norepinephrine (μg/24 h)	1.12	1.06–1.18	<0.001
Caffeine (μg/mL)	3.23	1.35–7.74	<0.01
Arousals (events/h)	1.31	1.17–1.46	<0.001

Data are presented as odds ratio (OR), interval of confidence of 95% (IC95%) and correspondent value of *p*; BMI, body-mass index; LDL, low-density lipoprotein; HDL, high-density lipoprotein; TG, triglycerides.

**Table 5 nutrients-14-01382-t005:** Multivariate logistic regression for four different models with OSA as dependent variable and testing the effect of caffeine on variables association with the development of the pathology. Variables for testing caffeine effect on the association of OSA with: (a) dysmetabolism-related comorbidities; (b) dyslipidaemia; (c) sympathetic nervous system (SNS) dysfunction; and (d) sleep fragmentation.

(a) Dysmetabolism model:			
DependentVariable = OSA Independent Variables	OR	IC_95%_	*p* Value
Caffeine (μg/mL)	3.05	0.587–15.87	0.185
BMI (kg m^−2^)	2.71	1.38–5.31	<0.01
Glycaemia (mg/dL)	1.234	1.03–1.48	<0.05
(b) Dyslipidaemia model:			
Dependent Variable = OSA Independent Variables	OR	IC_95%_	*p* Value
Caffeine (μg/mL)	6.93	0.79–61.18	0.081
TG (mg/dL)	1.01	0.99–1.02	0.304
HDL-c (mg/dL)	0.88	0.80–0.97	<0.01
LDL-c (mg/dL)	1.03	1.00–1.06	<0.05
(c) SNS model:			
Dependent Variable = OSA Independent Variables	OR	IC_95%_	*p* Value
Caffeine (μg/mL)	11.28	1.29–98.66	<0.05
Dopamine (μg/24 h)	1.00	0.98–1.01	0.263
Norepinephrine (μg/24 h)	1.14	1.06–1.23	<0.01
(d) Sleep model:			
Dependent Variable = OSA Independent Variables	OR	IC_95%_	*p* Value
Caffeine (μg/mL)	1.77	0.58–5.45	0.318
Arousals (events/h)	1.37	1.16–1.61	<0.001

## Data Availability

Data available on request due to restrictions (ethical reason). The data presented in this study are available on request from the corresponding author. The data are not publicly available due to ethical concerns.

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
