# Peer review of "Dysmetabolism and Sleep Fragmentation in Obstructive Sleep Apnea Patients Run Independently of High Caffeine Consumption"

_nutrients, 2022, doi:10.3390/nu14071382_

Round 1
Reviewer 1 Report
The study shows the association of caffeine with obstructive sleep apnea. It seems very well thought out to me. The study design is clearly presented. The tables supporting the results are well designed. The authors have hit on a very relevant topic by studying the effects of caffeine on sleep.
Author Response
thanks for your comments. we appreciated.
Reviewer 2 Report
After reading the article, below I present my comments and questions.
The research hypothesis is a bit doubtful because caffeine is considered a respiratory stimulator (e.g. it improves the work of respiratory muscles and widens the bronchi) and therefore caffeine citrate remains the gold standard in the treatment of apnea in premature newborns.
It is worth mentioning this in the introduction to the article.
In my opinion, the study is not well conducted in terms of methodology.
The authors included in the study persons with a very large age difference between the study group (mean age 50 years) and the control group (mean age 34 years).
With such a difference, it is not surprising that people with OSA had a significantly higher BMI and worse blood lipid parameters than healthy people, which may increase the risk of OSA. The strength of these differences in the multivariate analysis could therefore suppress the potential effect of caffeine.
Why did the authors analyze the effect of caffeine on sleep-related parameters in the (Non-OSA + OSA) and OSA patients group?
It is not known if the respondents knew before that they would take part in the study? This may have influenced caffeine consumption in the last day prior to the study, as the authors write doctors tend to recommend avoiding caffeine. Moreover, did the patients declare that such consumption of caffeine as just before the study is typical for them?
Why do the authors throughout the article write that they investigated the effect of coffee / caffeine consumption, when there is no data on this consumption in the study results? Conclusion also begins with the statement "Caffeine consumption in OSA patients is not associated with OSA severity ..."?
Caffeine is also not the same as coffee and it can come from different sources. It is also known that the type of coffee (filtered or unfiltered) affects blood lipid parameters.
According to the methodology of the study caffeine, theobromine, theophylline and paraxanthine were determined in the blood while there is only caffeine in the results?
Why the concentration of epinephrine in urine is not included in the analysis, since it was determined? What it comes from?
In my opinion, these are serious methodological doubts that make the conclusion of the study is debatable.
Author Response
We acknowledge the reviewer criticisms and hope that in the present version the manuscript will be suitable for publication. Please find below the answers to the reviewer comments/questions.
- The research hypothesis is a bit doubtful because caffeine is considered a respiratory stimulator (e.g. it improves the work of respiratory muscles and widens the bronchi) and therefore caffeine citrate remains the gold standard in the treatment of apnea in premature newborns. It is worth mentioning this in the introduction to the article.
It is not clear for the authors what is the link between the study described in the present manuscript and the role of caffeine on ventilation.
We know that caffeine, and in fact all methylxanthines, is considered a respiratory stimulator and decrease frequency of apneic episodes in premature infants (Steer et al., 2004) via A1 and A2A adenosine receptor inhibition in central respiratory neurons (Herlenius and Lagercrantz, 1999), making caffeine the drug of choice to treat apneas of prematurity (Mathew, 2011). However, in adult animal models, acute caffeine administration acts on A2A and A2B adenosine receptors inhibiting hypoxia-driven CSN activity by nearly 60%; A2B mediated inhibition is produced via interaction with dopamine metabolism in carotid body chemoreceptor cells while the A2A effect is postsynaptic (Conde et al., 2006; Conde et al., 2008). The immatureness of the carotid bodies in neonates (Gonzalez et al., 1994) explains the prevalence of the central stimulatory effect in newborns animals while in adults, with carotid body function fully expressed, the acute peripheral inhibitory effect of caffeine takes over and inhibition of ventilation is noticeable only when the drive of ventilation depends mostly on carotid body chemoreceptors, as it happens in hypoxia (Howell and Landrum, 1995). Moreover, we have shown, in the past, that caffeine blocks increased carotid body chemosensitivity in chronic intermittent hypoxic animals (Sacramento et al. 2015).
We can establish associations between caffeine consumption and ventilation and metabolism, although we believe that this is outside the scope of the present manuscript.
- In my opinion, the study is not well conducted in terms of methodology. The authors included in the study persons with a very large age difference between the study group (mean age 50 years) and the control group (mean age 34 years). With such a difference, it is not surprising that people with OSA had a significantly higher BMI and worse blood lipid parameters than healthy people, which may increase the risk of OSA. The strength of these differences in the multivariate analysis could therefore suppress the potential effect of caffeine.
We are aware that we have a difference in terms of age of individuals between OSA and non-OSA groups. However, we are also aware that it is not age that defines BMI or caffeine consumption. Here we were interested in evaluating the effect of caffeine on OSA severity and its comorbidities and not age or BMI on OSA.
- Why did the authors analyze the effect of caffeine on sleep-related parameters in the (Non-OSA + OSA) and OSA patients group?
The reviewer is right. it does not make any sense to analyze the effect of caffeine on NON OSA+OSA patients, as this does not add any valuable information. We have removed this from table 3 and from the text.
- It is not known if the respondents knew before that they would take part in the study? This may have influenced caffeine consumption in the last day prior to the study, as the authors write doctors tend to recommend avoiding caffeine. Moreover, did the patients declare that such consumption of caffeine as just before the study is typical for them?
No, patients did not knew about this particular study before attending the hospital appointment and in fact, the primary goal of the study was not related with the consumption of caffeine and OSA severity and impact on OSA-associated diseases. The information obtained from the patients included not only beverages but also food as well as medications.
- Why do the authors throughout the article write that they investigated the effect of coffee / caffeine consumption, when there is no data on this consumption in the study results? Conclusion also begins with the statement "Caffeine consumption in OSA patients is not associated with OSA severity ..."? Caffeine is also not the same as coffee and it can come from different sources. It is also known that the type of coffee (filtered or unfiltered) affects blood lipid parameters.
The reviewer is right. We know that caffeine can come from other sources apart from coffee and therefore we have replaced the terms coffee/caffeine consumption throughout the manuscript by caffeine consumption where appropriated. The only thing we can say is that caffeine levels are not associated with OSA severity or with OSA-associated dysmetabolic states.
- According to the methodology of the study caffeine, theobromine, theophylline and paraxanthine were determined in the blood while there is only caffeine in the results?
We have also assessed total xanthines in plasma, by quantifying, using HPLC, theobromine, theophylline and paraxanthine apart from caffeine. We have now included the total xanthines levels in table 2. However, our results apart from caffeine do not show significant differences in total xanthines between groups and therefore we only included caffeine in the statistical analysis.
- Why the concentration of epinephrine in urine is not included in the analysis, since it was determined? What it comes from?
As stated in lines 188-190 of the results section, we did not include variables without statistical significance in the binary logistic regression. As epinephrine levels between OSA and non-OSA, described in table 1, did not change significantly this variable was not included in the following statistical analysis.
Round 2
Reviewer 2 Report
After reading the corrected version of the article and the explanations of the authors, I do not submit any other comments.
Thank you.
Author Response
Thank you